# Predicting Conversion Rates in Online Hotel Bookings with Customer Reviews

**Liang Tang** [1,†], **Xi Wang** [2,*,†] and **Eojina Kim** [3]

1 Apparel, Events, and Hospitality Management, College of Human Sciences, Iowa State University, Ames, IA 50011, USA
2 Culture, Creativity and Management, School of Culture and Creativity, BNU-HKBU United International College, Zhuhai 519085, China
3 Hospitality and Tourism Management, Pamplin College of Business, Virginia Tech, Blacksburg, VA 24061, USA
* Correspondence: xiwang@uic.edu.cn
† The first author and second author contributed to the paper equivalently.

**Abstract:** E-commerce in the hospitality and tourism field has already ranked No. 2 among all online shopping categories worldwide. However, customers' visits to a hotel booking website cannot guarantee the generation of sales, while the conversion rate is regarded as the indicator that effectively assesses the e-commerce website performance. This study aimed to investigate the influential factors of conversion rates from both affective content and the communication style of customer's online reviews. The affective content was evaluated with eight emotional dimensions (i.e., joy, sadness, anger, fear, trust, disgust, anticipation, and surprise) in Plutchik's emotion wheel, and the communication style perspective was assessed with linguistic style matching (LSM). In total, 111,926 customer reviews from 641 hotels in five cities in the U.S. were collected for the analysis. Results indicated that LSM and four emotions have significant impacts on hotel conversion rates. This research contributes to the knowledge body of customers' conversion behaviors on hotel booking websites and offers pertinent practical implications.

**Keywords:** conversion rate; e-commerce in hospitality; hotel booking; Plutchik's emotion wheel; linguistic style matching

## 1. Introduction

The emergence of e-commerce has restructured marketing activities in the hospitality and tourism industry since the digital realms overhauled customers' purchase patterns [1,2]. Based on customers' claimed purchasing, e-commerce in the hospitality and tourism field ranked No. 2 among all online shopping categories worldwide in 2018 [3]. Specifically, hotel reservations contributed the most and accounted for 39% of all the e-bookings in the hospitality and tourism industry [4]. Similar to the e-retailers in other manufacturing and service industries, hoteliers have made tremendous marketing efforts by improving website traffic on e-commerce channels [5]. However, a big gap exists between the numbers of lookers and bookers on hotel booking websites [6,7]. Previous academic studies have mostly focused on the impact of the content and design of e-commerce websites on customers' behavioral intentions rather than actual booking patronage due to the challenges of data collection [8,9]. But behavioral intentions cannot exactly predict actual behavior [10]. Therefore, it is crucial to investigate how to convert lookers to actual bookers on e-commerce websites (rather than just having the intention of booking) [11]. The look-to-book ratio is termed as the conversion rate (CR), which is used to differentiate between the numbers of browsers on an e-commercial platform and the number of customers who actually buy online. CR is a key indicator of an e-commerce website's performance [12]. The average CR of hotel booking websites is only 2.2%, with the top 20% of hotels at 5.6% and

the bottom 20% of hotels at 0.3% [13]. Even a marginal increase in CR could lead to a considerable growth in sales [14]; thus, how to improve CR on booking websites is of the utmost important for hoteliers.

Despite the importance of CR, limited studies on this topic have been found in the general business's e-commerce discipline [15,16], and even fewer in the hospitality and tourism field. To the knowledge of the present authors, Cezar and Ogut [11] and Horan et al. [12] were the only two studies in this field. The variables in these two studies are exclusively "quantitative surrogates" on internet bookings (e.g., review rating and tourism volume). However, controversy exists around the predictive outcome of these numerical quality diagnostics on customers' attitudes and behaviors (e.g., assessment of review helpfulness, and conversion behavior) on e-commerce websites [17–19]. For example, previous studies identified the contrasting effect of the same "quantitative surrogate" on an attitudinal or behavioral variable (e.g., the influence of hotel star ratings on room sales is positive in Ye et al. [20] but negative in Ye et al. [21]). Thus, some scholars suggested that natural language reviews should be given more attention [22,23], as indicated by Xu [24] that "online customer textual reviews contain rich information about customers' consumption experience, which reflect their perceptions of the detailed attributes of product and service quality." The precise and emotion-charged information expressed by language cannot be replaced with a "quantitative surrogate" [25]. Since the enriched content of textual reviews boosts information and emotional communications between reviewers and readers, it shows a greater capability for predicting readers' feelings and actions [26].

In the same vein, the present authors explored CR as a measure of customers' conversion behavior from the perspective of textual reviews. Furthermore, Ludwig et al. [27] suggested that "new insights into the customer review phenomenon can be derived from studying the semantic content and style properties of verbatim customer reviews to examine their influence." Thus, the present study analyzed textual reviews from the two perspectives of semantic content and linguistic style.

A focal element of semantic content is emotion. Emotion depicts innate or intuitive moods in contrast with rationale or knowledge. Emotions in textual reviews have been widely investigated as an antecedent of customers' attitudes (e.g., satisfaction and review helpfulness) or behavioral intentions [25,28,29]. One-dimensional scalability (positive vs. negative) has been mostly adopted in previous studies [26,30]. However, Pfister and Bohm [31] argued that the multiple dimensions (e.g., anger and trust) underlying either positive or negative emotions have distinct origins, denotations, and appraisals. Plutchik's wheel is a four-dimensional circumplex with eight basic emotions, intensities, opposites, and combinations, which allow researchers to explore complicated emotional systems of customers in depth [25]. Furthermore, the eight basic emotional dimensions in Plutchik's wheel, as subjective reactions (e.g., trust), are expected to result in behavioral reactions (e.g., groom generated by trust), which further predict survival functions (e.g., mutual support generated by groom) [32]. The distinct survival functions initiated by subjective reactions (i.e., basic emotions) indicate that the eight basic emotions embedded in reviews may generate impacts on CR at distinct extents. Such information is crucial for industry practitioners to improve CR via strategically presenting reviews addressing different emotions (e.g., ranking and filter) on e-commerce websites. To the knowledge of the present authors, Ludwig et al. [27], as the only study that investigated the relationship between affection and CR, adopted the one-dimensional scalability of affection (i.e., positive vs. negative). To take one step further, the present study examined how CR is impacted by the eight aspects (i.e., joy, sadness, anger, fear, trust, disgust, anticipation, and surprise) of Plutchik's emotional wheel, which is a cornerstone emotional framework in psychology.

Besides semantic content, linguistic style features of online reviews generate substantial influences on readers' comprehension and perception [33]. Coordination is required in human communication. Each participant regulates his/her own presentation of ideas and opinions in the conversation to strengthen mutual attractions and improve shared comprehension [34]. One of the critical strategies for such a regulation is language mimicry.

Linguistic style matching (LSM) is defined as an algorithm to automatically evaluate language mimicry with the adoption of functional words [35]. According to the similarity-liking premise proposed by Byrne et al. [36], greater LSM potentially leads to a stronger bond between the parties, which is a predictor of their attitudes and behaviors. Therefore, LSM of the review bundle for a specific hotel was assessed as a diagnostic cue for CR (i.e., as a behavioral construct) in the current study.

The purpose of the present study is to investigate the influential factors of CR on an individual hotel booking website from the affective content (emotion) and communication style (LSM) aspects. The specific objectives of the research include: (1) assessing the impact of eight individual emotional dimensions (i.e., Plutchik's wheel) embedded in reviews on CR, and (2) examining the impact of LSM embedded in reviews on CR. The researchers selected www.booking.com as the investigation site since it is ranked as the No. 1 hospitality e-commerce and hotel booking website across the world by Bramblett [37]. Furthermore, this website offers real-time browsing and booking statuses for an individual hotel listing, which are required for counting CR. CR examined in the present study is a crucial indicator of website visitors' patronage behaviors, leading to hotels' revenue. Thus, our innovative assessment of CR's determinant factors from semantic content and linguistic style not only paves a new research venue, but it also generates far-reaching implications in the hospitality and tourism industry.

## 2. Literature Review

### 2.1. Conversion Rate and Information Processing Theory

The conversion rate (CR) on an e-commerce website is defined as the percentage of browsing on a website that ends with a purchase within a specific time window [38]. For example, if 100 visitors browse an e-commerce website and one of them places an order, the CR for the site is 1%. A high CR refers to a great percentage of customer patronages on the site on each visit, while a low CR means that most visitors browse the website without generating sales [15].

Previous studies on the topic of CR could be classified into four streams. The first stream examined the impact of website design on CR. Examples of website features assessed in the past research were composed of promotion (e.g., discount policy), quality (e.g., luxury products) [39], and usability factors [40]. The second stream was to establish the CR prediction mechanism with machine learning and other mathematical models by scholars in computer science [41,42]. The third stream was to address the importance of customers' previous e-shopping experience on CR. For example, Moe and Fader [16] tested how CR is influenced by customers' past visit experiences, whereas Li and Kannan [43] investigated the carryover and spillover effects of previous website visits on CR. The last and widely ignored stream was to examine how recommendation systems and textual reviews impact CR. The majority of previous studies in this stream focused on recommendation systems. For example, Ghose and Yang [44] and Rutz and Bucklin [45] confirmed the relationship between the rank order of a listing among customers and CR. To the knowledge of the present authors, very sporadic research investigated how textual reviews influence CR on an e-commerce website. The work of Ludwig et al. [27] is such an example with the study setting of www.Amazon.com.

The relationship between textual reviews and CR could be explained with information processing theory. Information processing theory describes an individual's learning process, which covers four steps including input, processing, storage, and output [46]. Humans receive input about our surroundings through our sensory organs (e.g., eyes and ears), perform analysis on information both cognitively and affectively, store information either in short-term or long-term memory, and present the results as an output [47]. Particularly, language as a means for expressing cultural values is crucial in the design and content of the messages that senders convey [48]. When receivers are familiar with the language used in the message, they are inclined to perceive less risk, identify higher usability, and generate greater satisfaction [49].

Language expressed either orally or textually (i.e., input in the information processing theory) has been widely proven in the psychology discipline to generate robust explanative and predictive impacts on an individual's behavior (i.e., output in the information processing theory), and such a linkage has been applied in different fields (e.g., education, marketing, and community service) to improve communication effectiveness [50–52]. In the same vein, textual reviews as the input of language have been proven to influence customers' behaviors (e.g., recommendations to others, patronage, and CR) as output in the information processing theory [53]. Specifically, Schindler and Bickart [54] suggested that textual reviews should be deciphered through the basic word categorizations of content (e.g., adjective) and communication styles (e.g., functional words). The justifications for including emotion (i.e., as one content component) and language style matching (i.e., as one communication-style component) as the determinant variables of CR are presented in the "Emotions" section and the "Linguistic Style Matching (LSM)" section.

## 2.2. Customer Review Emotion

Emotion is defined as a neural circuit, feedback mechanism, and feeling status that activates and structures cognitions and behaviors [55]. Emotion is either intrapersonal or interpersonal. The intrapersonal emotion depicts an individual's internal affections, whereas the interpersonal emotion represents the external detectable articulation of affections in the interactive communications [56]. The emotional representation in textual reviews emphasizes the interpersonal dimension [57].

Previous studies of the e-commerce in the hospitality and tourism field predominantly adopted a dichotomy of emotions: positive vs. negative [26,30,58]. An assumption behind these studies is that all emotional states are naturally sorted as either positive or negative. However, this presumption is not always tenable since ample evidence supports the idea that an individual's choice does not confirm to simple scalability [59]. By realizing the dichotomy's limitation, sporadic efforts were made to investigate more specific emotional dimensions in user-generated content [28,60,61]. The present study examines the eight dimensions of Plutchik's emotional wheel, with the rationale explained in the next subsection.

## 2.3. Multi-Dimensional Emotional Framework

A multi-dimensional approach has been advanced to feature discrete emotions in the psychology discipline [32,62,63]. Specifically, the Geneva emotional model (GEM) was proposed by Scherer et al. [63] GEM covers 10 positive dimensions and 10 negative ones. Fredrickson et al. [64] created a modified differential emotions scale (mDES). The mDES includes a 10-item positive emotion subscale and an 8-item negative emotion subscale. Ekman generated six fundamental emotion aspects: joy, anger, disgust, sadness, fear, and surprise [62]. Plutchik enriched Ekman's framework by adding two more emotional dimensions, adding trust and anticipation, and mapped the eight basic emotional aspects on a wheel [32]. The eight basic emotional aspects are revealed with four contrasting pairs: sadness–joy, fear–anger, disgust–trust, and surprise–anticipation. The word examples in each emotional dimension are shown in Table 1.

**Table 1.** Sample words of Plutchik's wheel.

| Emotional Dimensions | Sample Words |
| --- | --- |
| Anger | annoying, battle, complaint, dispute |
| Anticipation | attempt, countdown, develop, expected |
| Disgust | abject, contamination, defective, falsity |
| Fear | afraid, barbarian, cautionary, defend |
| Joy | abundant, beautiful, charmed, delicious |
| Sadness | abortive, badly, embarrass, fell |
| Surprise | amaze, occasional, quickness, rapid |
| Trust | accountable, believed, cohesive, deputy |

As shown in Figure 1, each emotional dimension belongs to either positive sentiment (represented with "+" category) or negative sentiment (represented with "−" category) has a different intensity, which is shown vertically on each spoke. Specifically, presented by the fluctuation of the color and saturation, the emotional intensity is indicating increases from the outer edge to the center. For example, anger serves as one of the eight emotional dimensions. A lower extent of anger is annoyance on the outer edge, whereas a greater extent of anger is rage in the center of the wheel. A dyad emotion sits between any two adjacent spokes, which rises out of the respective pairs of basic emotions. For example, anger and anticipation, as two foundational emotional dimensions, are mixed to generate aggressiveness. Plutchik also indicated that tertiary feelings exist that are a combination of three or more basic emotions, which are not shown in the emotional wheel [65].

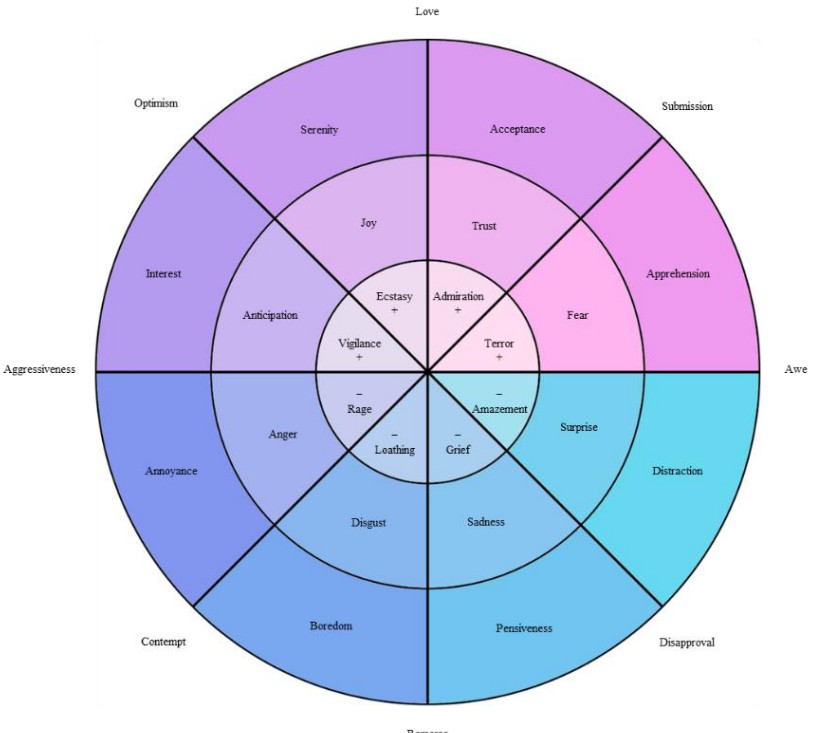

**Figure 1.** Emotional wheel (Revised based on Plutchik's emotional wheel).

In the context of e-commerce, the emotional representation in textual reviews emphasizes the interpersonal dimension [57]. Wang et al. suggested that "in the interpersonal communication context, affective cues provided by the communicator could evoke automatic affective reactions of the recipient, which need the recipient's few processing resources, occur rapidly, and assist in the formation of attitude and behavior" [25]. Therefore, the sentiment cue embedded in a review could transfer to the mood or feeling of a reader, which predicts his/her purchase behavior [27]. In the present context, emotion in a textual review is particularly used to predict conversion behavior (i.e., CR). Based on the previous discussions, we propose that the four emotional dimensions at the positive extreme of Plutchik's wheel have positive impacts on CR, while the four emotional dimensions at the negative extreme have negative impacts on CR within the e-commerce context.

**Hypothesis 1.** *Joy embedded in reviews has a positive impact on CR.*

**Hypothesis 2.** *Anger embedded in reviews has a positive impact on CR.*

**Hypothesis 3.** *Trust embedded in reviews has a positive impact on CR.*

**Hypothesis 4.** *Anticipation embedded in reviews has a positive impact on CR.*

**Hypothesis 5.** *Sadness embedded in reviews has a negative impact on CR.*

**Hypothesis 6.** *Fear embedded in reviews has a negative impact on CR.*

**Hypothesis 7.** *Disgust embedded in reviews has a negative impact on CR.*

**Hypothesis 8.** *Surprise embedded in reviews has a negative impact on CR.*

*2.4. Linguistic Style Matching (LSM)*

Communication accommodation theory (CAT) is a foundational psychology theory of interpersonal or intergroup communications [34]. Communication accommodation refers to adjustments or changes made in any type of communication (e.g., verbal, text, and posture) in order to gain approval from the receiver, enhance communication effectiveness, and establish a positive social identity [66]. The purpose of CAT is to suggest a communicative adjustment strategy for the sender and to predict how the receiver perceives, assesses, and responds to the sender [67]. A core strategy of CAT is convergence, which refers to adjusting the communicative behavior to be consistent with one and another [68].

In verbal and written communication, such a convergence is represented as LSM. Language style shows the features of language expressions under the assistance of subjective and objective considerations. Specifically, LSM suggests that an individual is inclined to match his/her linguistic style to pursue synchronization with other parties [33]. LSM is an algorithm designed for assessing the extent of synchronization between two parties as it relates to the use of functional words [35]. The nine categories and their corresponding example words are provided in Table 2.

**Table 2.** LSM categories and sample words.

| Category | Examples |
| --- | --- |
| Personal pronouns | I, he, she |
| Impersonal pronouns | anyone, everyone, other |
| Articles | a, an, the |
| Conjunctions | as, if, when |
| Prepositions | across, between, for |
| Auxiliary verbs | can, do, become |
| High-frequency adverbs | extremely, hence, indeed |
| Negations | don't, doesn't, no |
| Quantifiers | all, every, few |

Community members show interest in playing "so-called language games . . . in which members adhere to the collective's style of conversing to demonstrate their affiliation" [69]. An e-commerce website for a specific product category (e.g., books, and hotels) could be viewed as an online community of customers who share a common interest or goal [70]. Chung and Pennebaker suggested that high LSM in the reviews written by customers could promote harmony and form a common identity on the website [71]. Moreover, Liu and Liu [72] indicated the following: in an online environment, the process of reading reviews essentially is a language exchange process, just using another form of verbal communication. Therefore, the LSM will have an impact on the result of the exchange. The LSM, or degree of synchronization between two conversants in terms of their use of function words, also has behavioral implications (p. 571).

Many previous studies have proven the crucial role of LSM in textual communication between individuals in online settings [27,33]. For example, Wang et al. indicated that the LSM of customer online review has significant effects and shows the convergence in the restaurant industry, which assists readers' diagnosticity of the review [25]. To take the idea even further, we aimed to test how LSM impacts conversion behavior. Accordingly, the hypothesis was created below.

**Hypothesis 9.** *LSM embedded in reviews has a positive impact on CR.*

## 3. Methodology

### 3.1. Data Collection

Data were legally collected from www.booking.com with the Datatool scraping software, which followed the guidance of the Institutional Review Board at the authors' affiliated university. Data scraping was completed 14–29 January 2020. Previous studies on CR usually collected data for one week [73,74]. However, to obtain a more precise result, we collected data for 16 days. Data were collected in July because it is a peak season for traveling with a higher volume of both visiting number and booking number than other seasons.

The study sample included the hotels at six cities among the best tourism destinations in the U.S. [75]. The six cities across diverse regions (North, South, Northeast, Northwest, Southeast, and Southwest) included Chicago, Houston, New York City, San Francisco, Orlando, and Las Vegas. We further set two screening criteria. The first criterion was to only keep hotels among different property types (e.g., motels and B&Bs). The second criterion was to remove the default review with no actual content (i.e., a statement shown as "there are no comments available for this review"). After screening unqualified ones, 641 hotels and 111,926 corresponding customer reviews remained for further analysis. The number of hotels and corresponding customer reviews in each of the cities' studies were 96(14,413), 101(5,717), 107(18,119), 107(22,736), 159(22,010), and 71(28,931), respectively.

### 3.2. Variables

As for the dependent variable of hotel website's conversion rate, since the data were collected for 16 days, the average daily webpage visiting number and booking number for each hotel were used to compute CR. The original information of the booking number on www.booking.com only shows on a daily basis (i.e., "xx times of booking during past 24 h"). However, the online booking status is dynamic, which means the information of the booking times showing on the website is always changing. Therefore, consistent with the approaches used in Cezar and Ogut [11], we collected the booking number on a 2 h basis and then took the average of them within a 24 h period from 12:00 a.m. to 11:59 p.m. to represent the daily information of CR. The formula of CR is demonstrated below.

$$hotel\ conversion\ rate = \frac{hotel\ booking\ number}{hotel\ webpage\ visiting\ number} \quad (1)$$

As for the independent variables, both customer review emotional dimensions and LSM were computed based on the online review contents. Developed by Pennebaker, LIWC can help identify emotional and structural components in diverse textual contexts, such as online review content of e-commerce. LIWC proposes to calculate the percentage of each word, and then indicates a specific value score for a certain measured category [76].

Taking advantage of LIWC, the independent variable of LSM was calculated with the following four steps. In the first step, the LIWC was used to compute the score for each of the nine functional word categories. In the second step, the nine scores for the corresponding nine components of functional words generated by LIWC were further used in the following formula [35]. The formula is illustrated with the example of the functional word category "articles".

$$LSM_{art} = 1 - [(art_1 - art_G) \div (art_1 + art_G + 0.0001)] \quad (2)$$

In the formula, $art_1$ indicates the percentage of "articles" in a specific customer review, while $art_G$ represents the overall percentage of "articles" in the remaining reviews within the same group. Step three was to generate the LSM score of a customer review by calculating the average of all the individual LSM scores for the corresponding nine components of

functional words. The LSM score of an individual review ranged from 0 to 1. The higher LSM score means there is a greater extent of LSM [77].

Meanwhile, regarding the eight emotional dimensions, since the default dictionary of LIWC only covers three specific emotions (anxiety, anger, and sadness) [76], the Word–Emotion Association Lexicon (EmoLex) developed by the National Research Council Canada was further imported into LIWC. The EmoLex was used to identify the eight emotional dimensions of Plutchik's emotional wheel, and then calculate the value score of every individual emotional dimension related to the customer's online review contents [78].

Five control variables were directly extracted from the website, including distance from a hotel to the city center, hotel room rate, hotel star rating, risk-free cancellation policy, and high-demand status. The cities (i.e., Chicago, Houston, New York City, San Francisco, Orlando, and Las Vegas) were used as the dummy variable. In order to calculate the dependent variable of CR, we collected the webpage visiting number and booking number for each of the 641 hotels. Descriptive statistic results of all the variables—the dependent variable, the independent variables, and the control variables—are present in Table 3.

**Table 3.** Descriptive statistic results of the variables.

| Variable | Mean | Median | S.D. | Min | Max | Range |
|---|---|---|---|---|---|---|
| CR | 0.389 | 0.357 | 0.105 | 0.004 | 0.891 | 0.888 |
| Anger | 1.724 | 0.000 | 3.991 | 0.000 | 75.000 | 75.000 |
| Anticipation | 3.321 | 1.790 | 4.906 | 0.000 | 60.000 | 60.000 |
| Disgust | 1.617 | 0.000 | 4.234 | 0.000 | 75.000 | 75.000 |
| Fear | 1.074 | 0.000 | 3.051 | 0.000 | 75.000 | 75.000 |
| Joy | 4.904 | 3.030 | 6.465 | 0.000 | 80.000 | 80.000 |
| Sadness | 1.638 | 0.000 | 3.503 | 0.000 | 75.000 | 75.000 |
| Surprise | 1.672 | 0.000 | 3.722 | 0.000 | 50.000 | 50.000 |
| Trust | 5.203 | 3.450 | 6.458 | 0.000 | 80.000 | 80.000 |
| LSM | 0.501 | 0.530 | 0.236 | 0.010 | 0.940 | 0.930 |
| Distance from a hotel to the city center | 3.060 | 1.100 | 4.121 | 0.066 | 19.300 | 19.234 |
| Hotel room rate | 139.2876 | 129 | 87.2025 | 29 | 849 | 820 |
| Hotel star rating | 2.9576 | 3.0000 | 0.9338 | 0 | 5 | 5 |
| Risk-free cancellation policy | 0.6290 | 1.0000 | 0.4831 | 0 | 1 | 1 |
| High-demand status | 0.6546 | 1.0000 | 0.4755 | 0 | 1 | 1 |

*3.3. Beta Regression Analysis*

Beta regression was used for data analysis, considering CR as the dependent variable, ranging from 0 to 1. Although ordinary least squares (OLS) can be generally used in most regression analyses, beta regression is especially designed for modeling rate and proportion variables [79]. Compared to traditional OLS, beta regression can avoid generating a fit value beyond the lower and upper bounds between 0 and 1. Additionally, beta regression eliminates the inefficient effect of estimators caused by the standard error [11]. In beta regression, the coefficient associated with a variable refers to the change in log-odds of the target outcome (e.g., "success rate" or "retention rate"). To take the emotional dimension of anticipation as an example, if the estimated log-odds of anticipation is 0.02, the odds for CR should change by 1.0202 times (i.e., exp (0.02) = 1.0202). That is to say, with 1-unit increase in anticipation, the estimated odds of CR would increase by 2.02%.

## 4. Results

Multicollinearity effects were tested for the assumption of beta regression as step one. The correlations between all the variables were examined first. CR was correlated with all the independent and control variables, except LSM and anticipation. Since there are some multicollinearity effects, the variance influence factor (VIF) for all the variable pairs were checked. VIF ranged from 1.17 to 6.52, which were satisfactory rates. Overall, the

multicollinearity effects were insignificant in the model, which satisfied the prerequisite of beta regression.

A twofold beta regression analysis was conducted in the present study. In the first step, regression analysis was only applied to all control variables in order to have a clear understanding of shadow effects. According to Table 4, the hotel room rate, risk-free cancellation policy, and high-demand status control variable showed a positive influence on the dependent variable CR, whereas distance from a hotel to the city center, hotel star rating, city of Houston, and city of New York had negative influences on the dependent variable. The Pseudo R-squared of the control variable's regression model was 0.17.

In the second step, we explored the effect of all focal variables of interests in the present study. After including the independent variables of interest (i.e., LSM and eight emotional dimensions), the Pseudo R-squared increased from 0.17 to 0.52. It indicated that the model fit was significantly increased 35% after introducing the independent variables of customer review emotion and LSMs. With the $p < 0.01$ and the log-likelihood of 670.90, the overall model fit of the beta regression analysis was acceptable.

Referring to the details presented in Table 4, four out of the eight emotional dimensions showed significant impacts on CR. Specifically, anticipation ($\beta = 0.02$; $p = 0.03$) and trust ($\beta = 0.03$; $p = 0.04$) had positive influences on CR, which meant that with an increase of both anticipation and trust emotions expressed in the customer reviews, there would also be an increase of hotel CR. Anger ($\beta = -0.04$; $p < 0.01$) and disgust ($\beta = -0.02$; $p < 0.01$) had negative influences on CR, that is to say, with an increase in either anger or disgust emotion contained in customer reviews, the hotel CR would also suffer with a significant decrease. As it relates to the linguistic style of customer reviews, LSM ($\beta = 0.71$; $p = 0.01$) had a positive effect on CR, which indicated that with a greater extent of LSM, the hotel CR would increase.

The results of the control variables aligned with the previous literature [11]. Based on the results of the research model, the high-demand status of a hotel ($\beta = 0.17$; $p < 0.01$) had a significantly positive impact on CR. However, the distance from a hotel to the city center ($\beta = -0.01$; $p = 0.04$), hotel star rating ($\beta = -0.06$; $p = 0.01$), and hotel room rate ($\beta = -0.01$; $p = 0.04$) had significantly negative impacts on CR. No significant difference could be identified among cities. The current study also conducted a robustness check with OLS, which showed consistent results for the beta regression analysis.

**Table 4.** Beta regression result and comparison.

| Variable | Model with Only Control Variables | | | | | Research Model | | | | |
|---|---|---|---|---|---|---|---|---|---|---|
| | Estimate | Std. Error | z Value | Pr(>\|z\|) | | Estimate | Std. Error | z Value | Pr(>\|z\|) | |
| (Intercept) | −0.69 | 0.12 | −5.70 | 0.00 | *** | −0.70 | 0.18 | −3.92 | 0.00 | *** |
| Distance from a hotel to the city center | −0.02 | 0.01 | −2.67 | 0.01 | ** | −0.01 | 0.01 | −1.98 | 0.04 | ** |
| Hotel star rating | −0.10 | 0.03 | −3.32 | 0.00 | *** | −0.06 | 0.02 | −2.46 | 0.01 | ** |
| Hotel room rate | 0.00 | 0.00 | −3.63 | 0.00 | *** | −0.01 | 0.00 | −2.00 | 0.04 | ** |
| Risk-free cancellation policy | 0.25 | 0.05 | 4.61 | 0.00 | *** | 0.06 | 0.04 | 1.38 | 0.17 | |
| High-demand status | 0.34 | 0.06 | 6.08 | 0.00 | *** | 0.17 | 0.04 | 3.98 | 0.00 | *** |
| City_Houston | −0.23 | 0.11 | −2.08 | 0.04 | ** | −0.08 | 0.08 | −1.02 | 0.31 | |
| City_LasVegas | 0.04 | 0.10 | 0.39 | 0.69 | | 0.14 | 0.08 | 1.85 | 0.06 | |
| City_NewYorkCity | −0.18 | 0.09 | −1.95 | 0.05 | | −0.03 | 0.07 | −0.48 | 0.63 | |
| City_Orlando | −0.07 | 0.11 | −0.63 | 0.53 | | 0.07 | 0.08 | 0.81 | 0.42 | |
| City_SanFrancisco | −0.02 | 0.10 | −0.16 | 0.88 | | −0.05 | 0.07 | −0.69 | 0.49 | |
| LSM | | | | | | 0.71 | 0.26 | 2.78 | 0.01 | *** |
| Anger | | | | | | −0.04 | 0.01 | −5.57 | 0.00 | *** |
| Anticipation | | | | | | 0.02 | 0.01 | 2.17 | 0.03 | ** |
| Disgust | | | | | | −0.02 | 0.01 | −3.68 | 0.00 | *** |

**Table 4.** *Cont.*

| Variable | Model with Only Control Variables | | | | Research Model | | | | |
|---|---|---|---|---|---|---|---|---|---|
| | Estimate | Std. Error | z Value | Pr(>\|z\|) | Estimate | Std. Error | z Value | Pr(>\|z\|) | |
| Fear | | | | | 0.01 | 0.01 | 1.00 | 0.32 | |
| Joy | | | | | −0.02 | 0.02 | −1.18 | 0.24 | |
| Sadness | | | | | 0.01 | 0.01 | 1.19 | 0.24 | |
| Surprise | | | | | 0.01 | 0.02 | 0.55 | 0.58 | |
| Trust | | | | | 0.03 | 0.01 | 2.06 | 0.04 | ** |

** $p < 0.05$; *** $p < 0.01$; Research model's Pseudo R-squared increased from 0.17 to 0.52.

## 5. Discussion and Implications

### 5.1. Theoretical Implications

The present study contributes to the extant research on customer behavior by exploring the effect of semantic content and LSM on online booking website and e-commerce platform of the hospital and tourism industry; meanwhile, it extends the extant literature in four important ways. First, the present research is a pioneer that investigates the influence of review content on CR from the perspectives of semantic content and LSM. Previous studies have primarily investigated readers' attitudes and behavioral intentions as consequences of checking reviews. For example, Radojevic et al. examined customer reviews on www. TripAdvisor.com and confirmed that hotel attributes and customers' characteristics most powerfully influenced customer satisfaction [80]. However, the present study focuses on CR, as it is an actual behavioral outcome that needs more attention from scholars. While the study of Ludwig et al. was the only existing one on this topic, it examined the effect of only the dichotomy of emotions (i.e., positive vs. negative) and LSM on the CR of book sales on www.Amazon.com [27]. Our research took Ludwig et al.'s idea further and deeper, particularly in e-commerce context of the hospitality and tourism field. This analysis followed the classic CR calculation precure by aggregating transaction level data, and although, along with the calculation, there may exist some systemic errors, the analysis results statistically confirmed the dynamic influences of LSM and four of the eight emotions on CR. LSM showed a positive impact on CR in the present study, which is consistent with Ludwig et al. [27].

Established based on the information processing theory, the present study also made significant contributions of the academical applications of the information processing theory in the e-commerce field. In the analysis, besides stating a plain truth of influence of the language style and language structure from online reviews, it further provided an in-depth consideration on both cognitive and affective contents (LSM and emotions) of customer reviews. With the purpose of improving the effectiveness of the communication, the results of the present study consolidated the theorical foundation of the information processing theory and, in addition, verified the distinctive effects of the customer's language expression on their corresponding behavior intention in the business and communication contexts as well.

Third, the present study is one of the pioneer studies to examine the effect of how individual emotional dimensions influence CR. In particular, the present study extends the extant literature on emotions and customer behavior by exploring the effects of various aspects of emotions on CR. Among the eight individual emotional categories of Plutchik's framework [32], CR is positively influenced by anticipation and trust and negatively influenced by anger and disgust. The results indicated that individual emotional dimensions have distinct impacts on CR.

Specifically, anticipation conveys a reviewer's perception on the future. It simulates a forthcoming situation into the affective present as an individual "evolves" into the anticipated state [81]. Thus, the readers could generate resonance with a reviewer's aspiration of the hotel, which accordingly leads to CR. Trustworthiness is the foundation for readers to

use the information of reviews to guide their behavior [82]. The honest and responsible image of a hotel conveyed in reviews could strengthen readers' confidence of their future experience at the hotel. Accordingly, trust shows a positive influence on CR.

Anger and disgust are two emotional dimensions in reviews that are mostly used to depict service failures [25]. When service failure happens, reviewers may express anger and disgust in online reviews to depict the disappointing consumption experience [58]. It could be explained with prospect theory. Prospect theory indicates that individuals give more weight to perceived gains versus perceived losses [83]. The present study confirmed that anger and disgust, which depict service failures in reviews, discouraged prospective customers from making a purchase (i.e., CR). The results are consistent with Wang et al., who indicated that both anger and disgust expressed in reviews are helpful for the perceptions of readers [25].

Analysis conducted in the present study provided the in-depth understanding on the conversion rate of hotel booking website. For instance, according to the results, the emotion of anger was identified with more of an intensive effect than disgust, even though both of them were identified with the negative impacts. In the same manner, the trust emotion was identified with more of an intensive effect than anticipation. The beta regression specially selected for the data analysis targeting to analyze the rate/ratio variable with an upper and lower bounds between 0 and 1, which ensures an accurate evaluation of the regression outcome for those intendent variables of interests.

Joy and sadness do not show significant impacts on CR. The writings of joy and sadness from business experiences are easily exaggerated, or even fabricated as fake or spam reviews [25,28]. Thus, joy and sadness expressed in reviews may generate mixed and insignificant impacts on readers' decision making (i.e., CR). Fear shows an insignificant impact on CR for lodging reservations.

Words in the category of fear (e.g., terror, and firearms) are primarily relevant to physical and psychological safety. In extremely rare situations, a hotel locates itself in a dangerous community or area. In other words, safety is almost a "pre-assumption" for most of the hotels. Thus, fear as an expression of unsafety in reviews cannot arouse echoes among readers, which further shows an insignificant influence on CR. Another emotional dimension that has an insignificant impact on CR is surprise. Surprise depicts a psychological status caused by an unexpected event. By analyzing its sample words (e.g., occasional, sudden, and unexpected), surprise may indicate a neutral, pleasing, or unpleasing experience, which is different from travel plans [84]. Considering the mixed nature of the unexpected experience, it is reasonable that the surprise expressed in reviews insignificantly impacts readers' CR.

Fourth, the present study adds to the literature of emotions by shedding light on the interpersonal role of emotions. Previous studies on the role of emotions on customer behavior have focused on an intrapersonal dimension of emotion within an individual customer [58,85]. However, the interpersonal nature of emotion has been widely overlooked. In the view of the significance effect of the intrapersonal perspective in this study, these dimensions should be accounted for when analyzing customers' emotions. With the popularity of e-commerce, the interpersonal role of emotion has been strengthened [57] since the sentiment cues expressed in a review could ignite affective responses in readers and consequently influence their attitudes and behaviors [25,28]. The present study contributes to understanding the interpersonal role of emotion.

Last, beyond the influence of the website features in the context of customer reviews, the present study underscores the important role of textual reviews. The limited studies on CR [11,12] in the context of customer reviews have emphasized website features (e.g., utility), leaving textual reviews sporadically discovered. To bridge these gaps, we contribute to the emerging research body on CR by examining textual reviews from both the content and linguistic style perspectives. In the hospitality and tourism field, we show, for the first time, how individual emotional dimensions and LSM play roles cognitively and affectively in shaping CR.

*5.2. Practical Implications*

The findings from the present study also provide insightful implications for industry practitioners who aim to increase CR for online retailers since customers increasingly utilize online reviews in making their hotel-booking decisions. First, the online travel agencies wishing to increase CR may be motivated to understand the drivers of CR. Due to the rapid growth of user-generated content and increasing usage of online reviews across a wide variety of online travel agencies, it would be vital to identify the predictors of CR. This can be done by analyzing customer sentiment and LSM, using the vast amount of real-time data. The findings from the present study suggest that industry practitioners could develop guidance for reviewers, especially elite reviewers, to express their views, as the linguistic nature of reviews ultimately influences CR in the e-commerce.

Second, not all of the eight emotional dimensions in reviews are persuasive enough to influence customers' purchase decisions. Thus, online travel agencies are advised to develop the criteria for ranking reviews in the recommendation system, considering the individual emotional dimensions, especially those that have significant impacts on prospective customers' conversion behavior. Currently for most online travel agencies, one of the primary review ranking criteria is customer rating (i.e., numerical rating from high to low or from low to high). The emotional analysis of textual reviews related to the four specific dimensions (i.e., anticipation, trust, disgust, and anger) could be incorporated into mathematic models of deciding the review rankings together with the overall numerical rating of the business. The industry practitioners are advised to evaluate their performance with our model, which is capable of predicting CR by taking into account sentiments and language matching along with online hotel booking features.

Third, hotel practitioners are also advised to examine specific emotional dimensions embedded in reviews. Accordingly, they could detect "whether the customers are happy with, dissatisfied with, losing trust in, or angry with their product or a particular feature of the product" [78]. The analysis of affective content from customer reviews, especially those with any of the four emotional dimensions that predict future customers' conversion behavior, could offer hotels an "emotion-aware system" to understand customers' feelings, discover their reasonings, and recover from service failures. Furthermore, the result of the present study also gives industry practitioners guidance on how to choose endorsement/compliment messages on the official websites of hotels. Specifically, the positive reviews that convey anticipation and trust are more convincing than those that convey joy, since they influence prospective customers' patronage decisions.

Fourth, LSM could be viewed as the analysis of linguistic style under a microscope. It is a creative approach to decode the underlying thinking procedure that results in readers' conversion behavior. A unique advantage of utilizing LSM to predict CR is to eliminate non-disruptive and lesser biases by comparing with self-reporting (e.g., post-review surveys with readers). When a hotel conveys similar thinking styles in the communicative messages with its customers, customers are inclined to generate feelings of intimacy and psychological synchrony. Thus, we suggest that in hotels' marketing and promotional materials (e.g., responses to reviewers' comments on third-party e-commerce or review websites), industry practitioners should adopt the functional words and word patterns demonstrating high-extent LSM with reviews, which is expected to enhance persuasion effectiveness.

**Author Contributions:** Conceptualization, L.T.; Methodology, X.W.; Software, X.W.; Validation, L.T. and E.K.; Formal Analysis, X.W.; Investigation, L.T. and E.K.; Resources, X.W.; Data Curation, X.W.; Writing—Original Draft Preparation, L.T. and X.W.; Writing—Review and Editing, L.T. and X.W.; Visualization, X.W.; Supervision, L.T.; Project Administration, X.W.; Funding Acquisition, X.W. All authors have read and agreed to the published version of the manuscript.

**Funding:** This work was supported in part by Guangdong Higher Education Upgrading Plan (2021–2025) of "Rushing to the Top, Making Up Shortcomings and Strengthening Special Features" with No. of UICR0400031-21.

**Institutional Review Board Statement:** Not applicable.

**Informed Consent Statement:** Not applicable.

**Data Availability Statement:** Data used in this study can be obtained by contacting the corresponding author.

**Conflicts of Interest:** The authors declare no conflict of interest.

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
