# Peer review of "Predicting Conversion Rates in Online Hotel Bookings with Customer Reviews"

_jtaer, doi:10.3390/jtaer17040064_

Round 1

Reviewer 1 Report

I have one minor concern for the study.

The study aims to investigate the influential factors of conversion rates from both affective content and communication style of customer’s online reviews. In literature review, the study used information processing theory to explain the relationship between textual reviews and CR. However, the authors do not justify this theory in theoretical implications.

Author Response

Dear reviewer,

Thank you very much for your valuable suggestion.

We totally agree with you that it is essential to add some academic implications to the information processing theory. And one more paragraph has been added in 5.1. Theoretical Implications section. Please refer to Section 5.1, lines 122-31.

Reviewer 2 Report

Dear authors, congratulations for your paper it is a paper very interesting to read, with a good methodology and that covers an important topic. However I have a few comments that require clarification:

1. Customers look at reviews, of course the language and the emotions they get from reading can increase or decrease their CR, I agree. However, must customers only read the first 10-15 comments, so have you considered all the comments as equally important? only the last ones? What if the comments about a hotel change because the hotel has acted as a consequence of the comments. How do you measure it?

2. How can you control that there is no people that are visiting the same website many different times, I can visit and check the same information three times, finally I buy, for you conversion is 1/3 but really is 100%. It is not so unlikely that someone checks the booking com for a destinity more than one. Have you considered it?

3. What are adding to the academic literature the paper? it copuld be expected that anger or disgust have a negative beta. So what is added that was not known before'

4. The link between emotions and some words looks not clear to me. why do you link rapid with surprise? If I write the hotel answer to my request was rapid, this is not a surprise, maybe means that I trust the hotel. In the same way you link the word defend with fear. Some of these connections are very unclear.

Author Response

Dear reviewer,

Thank you very much for your valuable suggestions and essential concerns. Please refer to our responses to the comments as follows.

Comment 1:

Customers look at reviews, of course the language and the emotions they get from reading can increase or decrease their CR, I agree. However, must customers only read the first 10-15 comments, so have you considered all the comments as equally important? only the last ones?What if the comments about a hotel change because the hotel has acted as a consequence of the comments. How do you measure it?

Response 1:

Thank you very much for your essential concerns. Actually, as you mentioned, it is impossible for every individual customer to read all the comments and then make a decision. However, in this study, we propose this study based on a dynamic foundation process of viewing and booking process of the online travel agency’s website.

That is, for example, if someone would like to book a hotel on Sep 1, 2022, of course, as you mentioned, he or she only needed to review around one or two pages of comments before September 1, maybe between June to September. However, when treating it from a dynamic perspective, if someone would like to book a hotel on January 1, 2022, he or she would review some comments between October to December 2021. Therefore, it is the reason we considered more reviews in the data analysis process.

Comment 2:

How can you control that there is no people that are visiting the same website many different times, I can visit and check the same information three times, finally I buy, for you conversion is 1/3 but really is 100%. It is not so unlikely that someone checks the booking com for a destinity more than one. Have you considered it?

Response 2:

Thank you very much for this critical concern. It is true that someone would like to check the same hotel more than once. This could bring some systematic errors when measuring the conversion rate.

Based on the previous study, the conversion rate in the e-commerce context is classic to be calculated by aggregating transaction-level data (the dynamic business transaction situation). And introduced by Cezar & Ögüt (2016), as for online purchasing, the conversion rate is calculated by the number of purchases divide the number of visits.

However, we also agree that there would be some systematic errors in the calculation if ignoring the effects of multiple visits from the same customer, and one more discussion related to this issue has been added to the manuscript in Section 5.1, lines 16-21.

Reference:

  • Cezar, A., & Ögüt, H. (2016). Analyzing conversion rates in online hotel booking: The role of customer reviews, recommendations and rank order in search listings. International Journal of Contemporary Hospitality Management, 28(2), 286-304.

Comment 3:

What are adding to the academic literature the paper? it could be expected that anger or disgust have a negative beta. So what is added that was not known before'

Response 3:

As summarized in the literature review, although some previous literature explored topics of emotions and conversion rate within the e-commerce context, most of the study yield at the sentiment level (positive vs. negative sentiments), and few study step further to analyze the detailed emotion’s influence from eight dimensions.

As to the results, although anger and disgust had negative impacts while trust and anticipation had positive impacts were commonsensible, analysis conducted in this study on the one hand, statistically demonstrated such general knowledge, on the other hand, also provided some in-depth understanding of the conversion rate of the hotel booking website.

For instance, according to the results, the emotion of anger has been identified with more an intensive effect than disgust, even though both of them were identified with negative impacts. In the same manner, the trust emotion has been identified with more of an intensive effect than anticipation. The beta regression was specially selected for the data analysis targeting to analyze the rate/ratio variable with upper and lower bounds between 0 and 1, which ensures an accurate evaluation of the regression outcome for those independent variable of interest.

One more paragraph related to the discussion has been added in section 5.1, lines 55-62.

Comment 4:

The link between emotions and some words looks not clear to me. why do you link rapid with surprise? If I write the hotel answer to my request was rapid, this is not a surprise, maybe means that I trust the hotel. In the same way you link the word defend with fear. Some of these connections are very unclear.

Response 4:

The emotion analysis conducted in this study was based on the Natural Language Processing method and the emotional dictionary (EmoLex) introduced by Mohammad and Turney (2011) from National Research Council Canada. This emotion analysis process has been widely used in many situations such as speech, messages, book chapters, and online UGC (Lim et al.,2018; Mathur et al., 2020).

Besides, table 1 only provided some sample words representing the emotional dimension, however, those sample word is not an either-or decision. That is, it allows words belonging to more than one category of emotion. For instance, as indicated by the EmoLex, the word of rapid belongs to emotions of anger, fear, and surprise (Mohammad & Turney, 2011).

References:

  • Lim, K. H., Lee, K. E., Kendal, D., Rashidi, L., Naghizade, E., Winter, S., & Vasardani, M. (2018, April). The grass is greener on the other side: Understanding the effects of green spaces on Twitter user sentiments. In Companion Proceedings of the The Web Conference 2018 (pp. 275-282).
  • Mathur, A., Kubde, P., & Vaidya, S. (2020). Emotional analysis using twitter data during pandemic situation: Covid-19. In 2020 5th International Conference on Communication and Electronics Systems (ICCES) (pp. 845-848). IEEE.
  • Mohammad, S.M. and Turney, P. (2022). Retrieved from https://www.saifmohammad.com/WebPages/NRC-Emotion-Lexicon.htm

Round 2

Reviewer 2 Report

Authors have considered my suggestions